# A network-based approach to integrate nutrient microenvironment in the prediction of synthetic lethality in cancer metabolism

Iñigo Apaolaza[1,2,3]☯, Edurne San José-Enériz[4,5,6]☯, Luis V. Valcarcel[1,4,5], Xabier Agirre[4,5,6], Felipe Prosper[4,5,6,7], Francisco J. Planes[1,2,3]*

1 Universidad de Navarra, Tecnun Escuela de Ingeniería, San Sebastián, Spain, 2 Universidad de Navarra, Centro de Ingeniería Biomédica, Pamplona, Spain, 3 Universidad de Navarra, DATAI Instituto de Ciencia de los Datos e Inteligencia Artificial, Pamplona, Spain, 4 Universidad de Navarra, CIMA Centro de Investigación de Medicina Aplicada, Pamplona, Spain, 5 IdiSNA Instituto de Investigación Sanitaria de Navarra, Pamplona, Spain, 6 CIBERONC Centro de Investigación Biomédica en Red de Cáncer, Pamplona, Spain, 7 Clínica Universidad de Navarra, Pamplona, Spain

☯ These authors contributed equally to this work.
* fplanes@tecnun.es

**Data Availability Statement:** All data are in the manuscript and in supporting information files.

## Abstract

Synthetic Lethality (SL) is currently defined as a type of genetic interaction in which the loss of function of either of two genes individually has limited effect in cell viability but inactivation of both genes simultaneously leads to cell death. Given the profound genomic aberrations acquired by tumor cells, which can be systematically identified with -omics data, SL is a promising concept in cancer research. In particular, SL has received much attention in the area of cancer metabolism, due to the fact that relevant functional alterations concentrate on key metabolic pathways that promote cellular proliferation. With the extensive prior knowledge about human metabolic networks, a number of computational methods have been developed to predict SL in cancer metabolism, including the genetic Minimal Cut Sets (gMCSs) approach. A major challenge in the application of SL approaches to cancer metabolism is to systematically integrate tumor microenvironment, given that genetic interactions and nutritional availability are interconnected to support proliferation. Here, we propose a more general definition of SL for cancer metabolism that combines genetic and environmental interactions, namely loss of gene functions and absence of nutrients in the environment. We extend our gMCSs approach to determine this new family of metabolic synthetic lethal interactions. A computational and experimental proof-of-concept is presented for predicting the lethality of dihydrofolate reductase (DHFR) inhibition in different environments. Finally, our approach is applied to identify extracellular nutrient dependences of tumor cells, elucidating cholesterol and myo-inositol depletion as potential vulnerabilities in different malignancies.

## Author summary

Metabolic reprogramming is one of the hallmarks of tumor cells. Synthetic lethality (SL) is a promising approach to exploit these metabolic alterations and elucidate cancer-

**Funding:** This work was supported by the Minister of Economy and Competitiveness of Spain [PID2019-110344RB-I00], PIBA Programme of the Basque Government [PIBA_2020_01_0055], Elkartek programme of the Basque Government [KK-2020/00008], Fundación Ramon Areces [PREMMAM] to F.J.P.; Instituto de Salud Carlos III (ISCIII) [PI16/02024, PI17/00701, PI19/01352, PI20/01306], CIBERONC (Co-financed with European Union FEDER funds) [CB16/12/00489], ERANET program ERAPerMed [MEET-AML], MINECO Explora [RTHALMY], Cancer Research UK and AECC under the Accelerator Award Programme [C355/A26819]. to F.P. and Instituto de Salud Carlos III (ISCIII) [FI17/00297] to L.V.V. L.V.V. received his salary from Instituto de Salud Carlos III (ISCIII). The funders had no role in study design, data collection and analysis, decision to publish, or preparation of the manuscript.

**Competing interests:** The authors declare no competing interests.

specific genetic dependences. However, current SL approaches do not systematically consider tumor microenvironment, which is particularly important in cancer metabolism in order to generalize identified genetic dependences. In this article, we directly address this issue and propose a more general approach to SL that integrates both genetic and environmental context of tumor cells. Our definition can help to contextualize genetic dependencies in different environmental scenarios, but it could also reveal nutrient dependencies according to the genetic context. We also provide a computational pipeline to identify this new family of synthetic lethals in genome-scale metabolic networks. A computational and experimental proof-of-concept is presented for predicting the lethality of dihydrofolate reductase (DHFR) inhibition in different environments. Finally, our approach is applied to identify extracellular nutrient dependences of cancer cell lines, elucidating cholesterol and myo-Inositol depletion as potential vulnerabilities in different malignancies.

## Introduction

The main challenge of precision oncology is to be able to translate accumulating–omics data into actionable treatments, personalized for individual patients [1]. Synthetic Lethality (SL), defined as a type of genetic interaction where the co-occurrence of two (or more) genetic events results in cellular death, while the occurrence of either event on its own is compatible with cell viability, represents a promising approach [2]. Given the underlying genetic variations in tumor cells, SL largely expands the number of possible drug targets and creates an opportunity for more selective therapies [3].

Extensive work has been done to predict SL in cancer using both experimental and computational approaches [4–7]. These approaches have been mainly driven by the availability of large-scale gene knockout screening data for an increasing number of cancer cell lines [8,9]. Importantly, they provide an experimental *in vitro* measure of cancer gene essentiality, which can be integrated with genomic and transcriptomic data in order to hypothesize SL and identify response biomarkers.

Cancer metabolism is an ideal target to exploit the concept of synthetic lethality. Metabolic reprogramming of tumor cells leads to phenotypes that are substantially different from the ones observed in their healthy counterpart cells, and that can be potentially used to elucidate novel therapeutic strategies [10]. Outstanding works exploiting SL in cancer metabolism are reported in the literature [11–13], where different vulnerabilities were identified according to the underlying genetic context found in tumor cells.

We previously developed a computational framework to predict synthetic lethality in cancer metabolism based on the concept of genetic Minimal Cut Sets (gMCSs) [14]. Given a reference human genome-scale metabolic network, such as Recon3D [15], gMCSs define minimal subsets of genes whose simultaneous loss block a particular metabolic target, in our case metabolites that are essential for cellular growth, *e.g.* nucleotides for DNA, amino acids for protein, lipids for cell membranes, etc. gMCSs generalize the concept of SL to genetic interactions of more than 2 genes. Using gene expression data as a proxy for the activity of metabolic enzymes, identified gMCSs were used to predict metabolic vulnerabilities in cancer. Our in-silico (network-based) approach was validated using large-scale *in vitro* gene-knockout screening data and *in vitro* functional studies in multiple myeloma.

Despite these promising results, the application of SL to cancer metabolism has still different challenges. One of them is the integration of tumor microenvironment in the prediction of SL [16], since metabolic genetic interactions and nutritional availability are intertwined to

support cellular proliferation. On the one hand, the presence/deprivation of certain nutrients in the environment could modify the metabolic landscape and explain tumor resistance/sensitivity to metabolic targets. For example, it was recently reported that the supplementation of uridine rescues the anti-leukemic effect of dihydroorotate dehydrogenase inhibition [17]. On the other hand, targeting nutrient dependencies of tumor cells is an emerging topic in cancer research [18], and they are directly connected to the genetic variations of tumor cells but disconnected from the synthetic lethality literature. A paradigmatic example is the dependency on extracellular L-asparagine in samples with loss of asparagine synthetase (Lazarus et al., 1969). In fact, L-asparagine depletion via asparaginase administration is an approved therapy for acute lymphoblastic leukemia [19].

These results evidence the need of a more general approach to target cancer metabolism that integrates both genetic and environmental factors. However, this has not been systematically explored in current approaches to SL in cancer metabolism. In this article, we propose a new definition of SL that that combines loss of gene functions and absence of nutrients in the environment. Our definition can help to contextualize genetic dependencies in different environmental scenarios, *e.g.* if extracellular uridine is absent in the microenvironment, then dihydroorotate dehydrogenase becomes lethal in leukemia. Moreover, we could reveal nutrient dependencies according to the genetic context, as the case mentioned above where the blockage of the uptake of L-asparagine is lethal in patients with loss of function of asparagine synthetase.

In the light of the above definition, we extend the gMCSs approach to search for this family of synthetic lethal interactions. A computational and experimental proof-of-concept is presented for predicting the lethality in different environments of a well-studied drug target in cancer metabolism, dihydrofolate reductase (DHFR) [20]. Finally, our novel approach is applied to predict extracellular nutrient dependences of *in vitro* cancer cell lines. We identified cholesterol and myo-inositol depletion as promising vulnerabilities of different tumors.

## Results

### Tumor nutrient environment and synthetic lethality

Fig 1 shows an example metabolic network under two different culture mediums (environmental contexts), CM1 and CM2. The network ultimately produces the metabolite C, which is essential to tumor growth (Fig 1A). Under CM1, $g_1$ and $g_2$ form a synthetic lethal pair because their simultaneous inactivation blocks biomass production while individual inactivation do not (Fig 1B). Note here that, if we consider a genetic context where $g_2$ is not active (due to low expression, deletion or loss-of-function mutation, for example), $g_1$ becomes an essential gene in this given context (Fig 1C).

On the other hand, when trying to translate these findings to the network in CM2, due to the additional presence of M2 with respect to CM1 (Fig 1D), the simultaneous knockout of the two genes ($g_1$ and $g_2$) is not lethal anymore (Fig 1E). This example illustrates that SL is dependent on the environmental context and a more general definition of SL is necessary in cancer metabolism in order to consider it together with the underlying genetic context. Here, we propose to identify synthetic lethal interactions involving loss of gene functions but also nutrients in the environment.

In our example, Fig 1F shows an example of synthetic lethal involving two genes and one nutrient: {$g_1$, $g_2$, *M2*}, *i.e.* the lack of activity of $g_1$ and $g_2$ and the absence of nutrient *M2* leads to cellular death. In a particular context where *M2* is not present in the environment and $g_2$ is not active, $g_1$ remains essential. Similarly, in other context where $g_1$ and $g_2$ are not active, the depletion of *M2* is lethal, *i.e.* cellular growth is dependent on *M2* availability. Thus, this

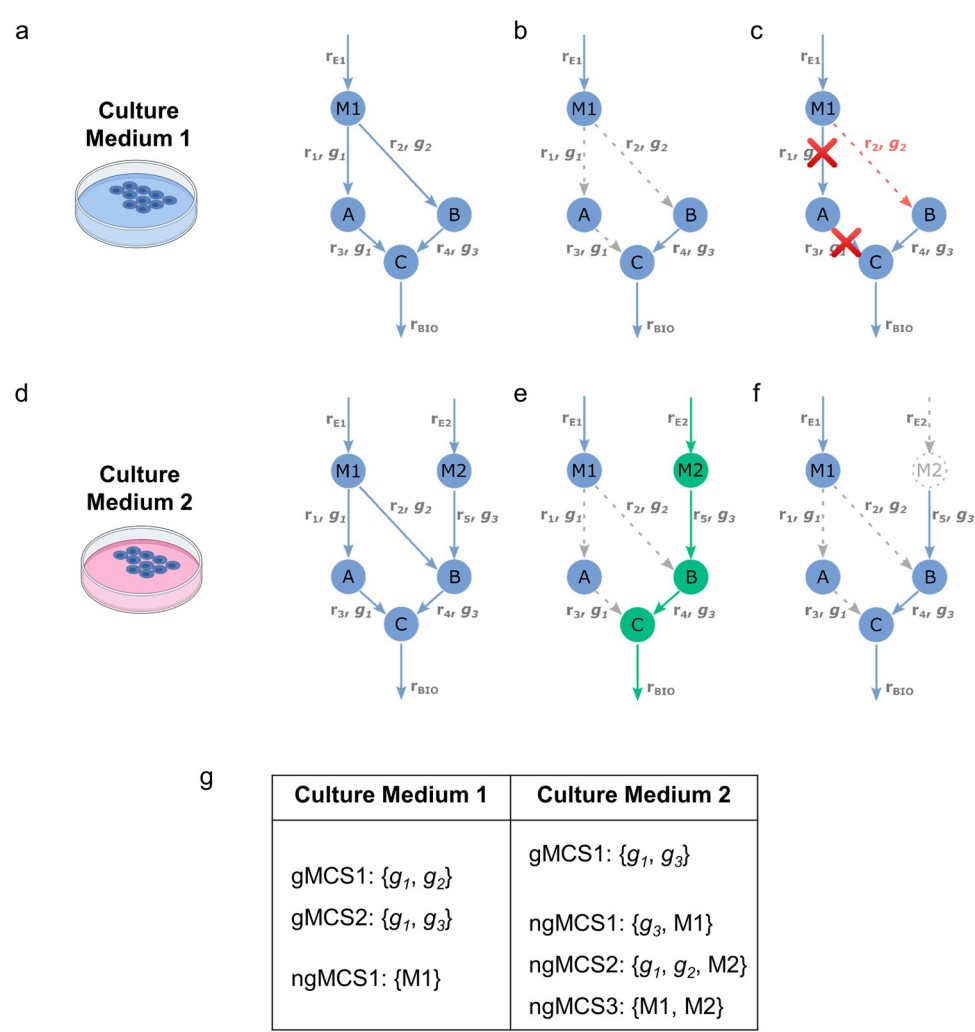

**Fig 1. Synthetic Lethality in 2 different environmental contexts. (a)** Example metabolic network under Culture Medium 1 (CM1). We have 5 reactions ($r_{E1}$, $r_1$, $r_2$, $r_3$, $r_4$), 4 metabolites (M1, A, B, C) and 3 genes ($g_1$, $g_2$, $g_3$). $r_{E1}$ is an input exchange reaction and represents the availability of M1; **(b)** $g_1$ and $g_2$ are synthetic lethals under CM1. **(c)** If $g_2$ is not active (red dotted line), the inhibition of $g_1$ is essential under CM1. **(d)** Example metabolic network under Culture Medium 2 (CM2). With respect to the metabolic network in (a), we have 2 additional reactions ($r_{E2}$, $r_5$) and 1 additional metabolite (M2). $r_{E2}$ is an input exchange reaction and represents the availability of M2; **(e)** g1 and g2 are not synthetic lethal under CM2 due to the alternative pathway via M2 degradation (in green). **(f)** The inhibition of g1 and g2 and deprivation of M2 in the metabolic network from (d) renders cellular proliferation impossible. **(g)** gMCSs and ngMCSs for CM1 and CM2. Gray dotted lines stand for not active/not present reactions/metabolites.

definition of SL is more general and allows us to identify both genetic and extracellular nutrient dependencies of tumor cells.

## Extending genetic MCSs into nutrient-genetic MCSs

In order to be able to systematically identify this novel family of synthetic lethals, we extend our previous network-based approach, based on genetic Minimal Cut Sets (gMCSs), leading to the concept of nutrient-genetic Minimal Cut Sets (ngMCSs). Here, we first revise the concept of gMCSs. Then, we introduce the concept of ngMCSs and describe how our approach is adapted for their calculation.

In the context of genome-scale metabolic networks, gMCSs were defined as minimal subsets of gene functions whose simultaneous loss block a particular metabolic task [14]. In cancer studies, we search for gMCSs that block the flux through the biomass reaction, an artificial equation that represents the metabolic biosynthetic requirements to support proliferation. At the metabolic level, gMCSs generalize the concept of SL to more than 2 genes.

With our new concept of ngMCSs, we aim to incorporate nutrient deprivations as part of the predicted synthetic lethal interactions. More specifically, ngMCSs are defined here as minimal subsets of gene functions and/or nutrients in the environment whose simultaneous loss block a particular metabolic task, here the flux through the biomass reaction. Fig 1G show the list of gMCSs and ngMCSs for the 2 environments considered. ngMCSs are directly connected with the type of synthetic lethals mentioned above that integrate genetic and environmental events.

While the loss of gene functions is modelled in genome-scale metabolic networks as gene knockouts, the loss of nutrients in the environment can be modelled as the knockout of input exchange reactions, which represent the uptake of nutrients. In Fig 1, $r_{E1}$ and $r_{E2}$ are the input exchange reactions associated with the uptake of *M1* and *M2*, respectively. Thus, ngMCSs define minimal subsets of gene knockouts and/or input exchange reaction knockouts that block the biomass reaction. For example, ngMCS2:{$g_1$, $g_2$, *M2*} involves the gene knockouts of $g_1$ and $g_2$ and the reaction knockout of $r_{E2}$, which is equivalent to the deprivation of *M2*. Our previously developed tool for the calculation of gMCSs, implemented in the COBRA toolbox [21], was amended to include the knockout of input exchange reactions in the solution space of gene knockouts and calculate ngMCSs (see Methods section and S1 Code).

Note here that both gMCSs and ngMCSs are structural properties of the reference network under specific growth medium conditions. However, as it was previously done with gMCSs [14], we can map experimental (-omics) data on identified ngMCSs and elucidate context-specific genetic or environmental dependencies (essential genes/nutrients). In other words, a gene or nutrient is essential for a particular context if it is the only active element in at least one ngMCS. This was illustrated above with the SL shown in Fig 1F, now ngMCS2, where if $g_1$ and $g_2$ are inactive in a particular context, *M1* becomes an essential nutrient; similarly, if, in a different context, $g_2$ is inactive and *M1* is absent in the environment, $g_1$ becomes an essential gene.

## Lethality of DHFR inhibition in different environments

As a proof-of-concept of our ngMCSs approach, we investigated a well-known metabolic target in cancer: dihydrofolate reductase (DHFR) [22]. The inhibition of DHFR has been proven lethal in different cancer cell lines under standard growth medium conditions [23]. Similarly, when we applied our gMCS approach to Recon3D [15], a high-quality reference human genome-scale metabolic network, under RPMI growth medium conditions, the same result regarding DHFR essentiality was obtained (see Methods section). On the other hand, in a more complex tumor microenvironment where all input nutrients annotated in Recon3D were available, we determined 17 gMCSs and 291 ngMCSs involving DHFR (S1 and S2 Tables). We identified 2 ngMCSs that involve DHFR and exclusively nutrients not included in standard RPMI growth medium (ngMCS5 and ngMCS289 detailed in S2 Table). For illustration, one of them involves {DHFR, thymidine, dihydrothymine, thymine}. Since thymidine, dihydrothymine and thymine are not part of the RPMI growth medium, this ngMCS indicates the essentiality of DHFR under these *in vitro* conditions and support our ngMCS approach in a general context.

Among the list of ngMCSs mentioned above, we identified 2 nutrients which have been previously associated with resistance to DHFR inhibition, namely, thymidine and

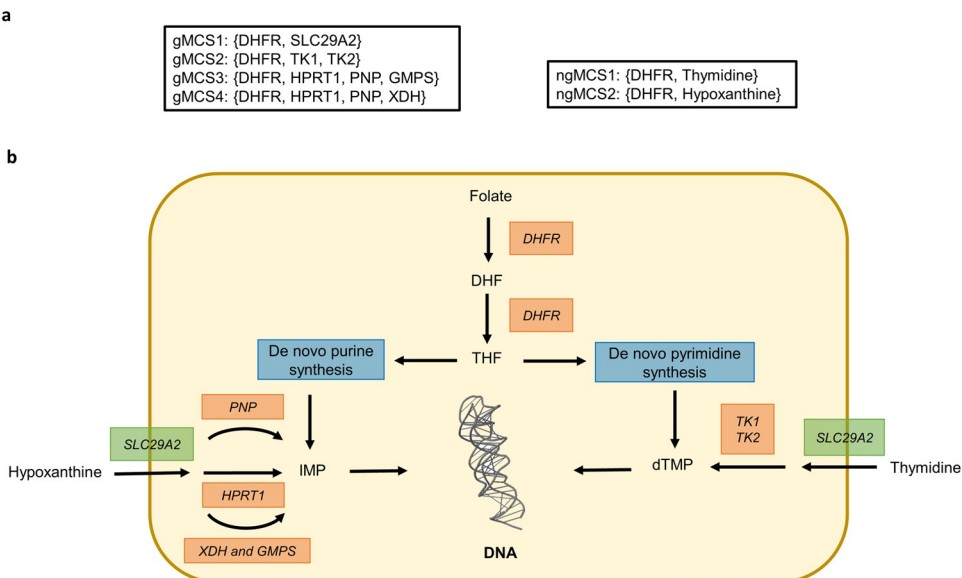

**Fig 2. Predicted synthetic lethals involving DHFR with thymidine and hypoxanthine in the growth medium. (a)** 4 genetic Minimal Cut Sets (gMCSs) and 2 nutrient-genetic Minimal Cut Sets (ngMCSs) involving DHFR. They were derived from Recon3D under the RPMI1640 growth medium plus thymidine and hypoxanthine; (**b**) Simplified network of metabolites and enzymes implied in the synthesis of purines and pyrimidines, emphasizing the role of DHFR, Thymidine and Hypoxanthine. Abbreviations: DHF: dihydrofolate; THF: tetrahydrofolate; IMP: inosinic acid; dTMP: 5-Thymidylic acid; DHFR: dihydrofolate reductase; PNP: purine nucleoside phosphorylase; HPRT1: hypoxanthine phosphoribosyltransferase 1; XDH: xanthine dehydrogenase; GMPS: Guanine Monophosphate Synthase; TK1: thymidine kinase 1; TK2: thymidine kinase 2; SLC29A2: solute carrier family 29 member 2.

hypoxanthine [23]. In fact, both metabolites were measured in different human biofluids [24], finding a relevant average plasma concentration for thymidine and hypoxanthine among cancer patients [25]. We investigated further this resistance mechanism with our approach and we re-calculated gMCSs and ngMCSs under the RPMI growth medium plus thymidine and hypoxanthine. Under this scenario, the list of gMCSs and ngMCSs involving DHFR is shown in Fig 2A. We have 2 ngMCSs, namely {DHFR, Thymidine} and {DHFR, Hypoxanthine}, which implies that the deprivation of either thymidine or hypoxanthine makes DHFR essential, as observed under standard RPMI growth medium. From a different angle, since DHFR is required in the de *novo* synthesis pathway of purines and pyrimidines, both thymidine and hypoxanthine are required to rescue proliferation upon DHFR knockout (Fig 2B). In addition, we have 4 gMCSs (Fig 2A), which indicate that the alternative salvage pathways through thymidine and hypoxanthine requires 1) the presence of the transporter SLC29A2 and 2) key enzymes, namely TK1 or TK2 for thymidine degradation, while HPRT1 or PNP or (GMPS and XDH) for hypoxanthine degradation (Fig 2B).

To validate our hypothesis, we conducted experimental study in 3 different cancer cell lines: JVM2, HT29 and PF382. Note here that, in contrast with PF-382 and JVM-2, HT-29 cells were not grown under RPMI but McCoy's 5a growth medium (see Methods section). The computational results shown above for DHFR under RPMI medium were the same for McCoy's 5a medium (S3–S5 Tables). We used Methotrexate (MTX) for selective inhibition of *DHFR*. We first calculated the MTX GI50 for these cell lines and validated their sensitivity (IC50 < 50 nM) (Fig 3A). Second, we showed that the decrease in proliferation mediated by MTX is rescued when both thymidine and hypoxanthine are added into the growth medium (Fig 3B, one-tailed Wilcoxon test p-value ≤ 0.05 for all cell lines), in line with our computational predictions described above. A similar result was found when thymidine and

Nutrient microenvironment and synthetic lethality in cancer metabolism

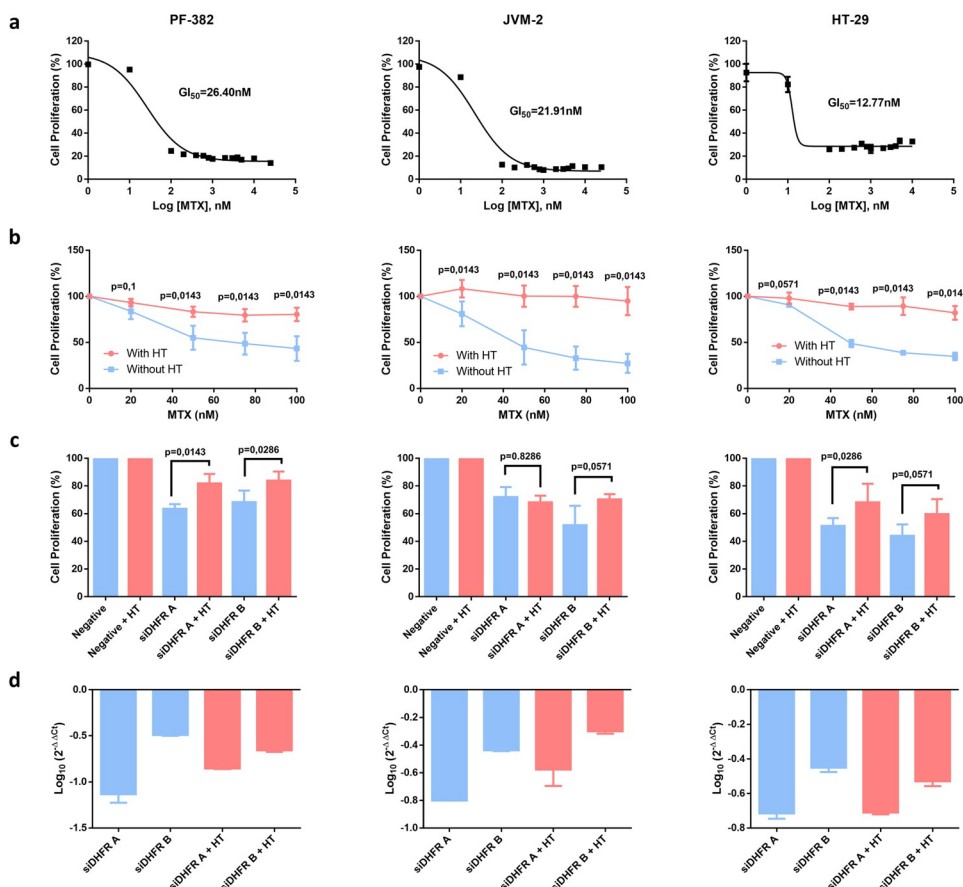

**Fig 3. In-vitro experimental validation of DHFR inhibition with hypoxanthine and thymidine in the growth medium. a)** GI50 values of MTX for PF-382, JVM-2 and HT-29 cell lines. **b)** Proliferation of PF-382, JVM-2 and HT-29 cell lines treated with different doses of MTX for 96h in presence or absence of Hypoxanthine-Thymidine. Data represent mean ± standard deviation of four experiments. **c)** Proliferation of PF-382, JVM-2 and HT-29 cell lines nucleofected with siRNAs targeted to *DHFR* gene in presence or absence of Hypoxanthine-Thymidine was studied by MTS at day 6 after nucleofection. The proliferation percentage refers to cells nucleofected with a negative control siRNA. Data represent mean ± standard deviation of at least three experiments. **d)** mRNA expression of *DHFR* gene 48 h after nucleofection with the specific siRNAs. Data are referred to GUS gene and an experimental group nucleofected with negative control siRNA. Data represent mean ± standard deviation of four experiments. All p-values (p) were determined by one-tailed Wilcoxon tests. Abbreviations: MTX: methotrexate; HT: Hypoxanthine-Thymidine.

hypoxanthine was added following *DHFR* silencing (Fig 3C), except for one of the siRNAs in JVM2 where no rescue of the proliferation decrease was observed. In particular, we found statistical significance in both siRNAs in PF-382 and one siRNA in HT29 (one-tailed Wilcoxon test p-value $\leq$ 0.05), while we were very close to statistical significance in one siRNA in HT29 and JVM-2 (one-tailed Wilcoxon test p-value: 0.0571). Full details of the experiment of *DHFR* silencing at different days can be found in S1 Fig. Note here that both *DHFR* siRNAs efficiently decreased DHFR expression in the three cell lines analyzed as detected by qRT-PCR (Fig 3D). Finally, coherent and high gene expression of *SLC29A2*, *TK1* and *HPRT1* was observed in the three cell lines considered, according to CCLE (Cancer Cell Line Encyclopedia) [9] (see S6 Table), which enable alternative salvage pathways for purine and pyrimidine synthesis through thymidine and hypoxanthine, respectively.

Considering the experimental data presented above, our ngMCSs approach was successful in determining in which environmental contexts *DHFR* inhibition is lethal in cancer. Based on

our computational approach, the opposite case was explored below, *i.e.* which genetic context makes lethal an extracellular nutrient perturbation.

## Extracellular nutrient dependences of tumor cells

Our ngMCS approach can also be used to systematically identify context-specific nutrient dependences of tumor cells, *i.e.* supply of extracellular metabolites that are essential for tumor proliferation in a particular genetic context. To illustrate this, we searched for ngMCSs in Recon3D under RPMI growth medium conditions (see Methods section). We identified 52 ngMCSs (S7 Table), which involve 19 nutrients. Neglecting essential nutrients under RPMI growth medium and ngMCSs involving more than one nutrient, we have 8 nutrients: L-asparagine, L-arginine, cholesterol, choline, D-glucose, L-glutamine, myo-Inositol and L-tyrosine. We discarded for further analysis choline and D-glucose because the associated genes in their ngMCSs show consistent and high expression and context-specific insights were not obtained (S2B and S2C Fig). The opposite occurs with L-tyrosine, whose associated genes in the ngMCSs are lowly expressed in most cell lines, which makes it an essential nutrient in most cases (S2A Fig). Thus, we focus on the other 5 nutrients (Fig 4A).

The dependency on extracellular L-asparagine in samples with loss of asparagine synthetase (ASNS) has been long studied in cancer research [26]. In fact, L-asparagine depletion via asparaginase is an approved therapeutic strategy for acute lymphoblastic leukemia, and it is being investigated in solid tumors [18]. The dependence on extracellular L-arginine and L-glutamine in samples with loss of glutamine synthetase (GLUL) and Argininosuccinate Synthase 1 (ASS1), respectively, has been validated with *in vitro* experiments of nutrient depletion and barcode genetic screens [18]. The clinical importance of the uptake of L-arginine and L-glutamine in different tumors has received much attention in the last years [27,28].

In the case of L-arginine, we also identified the role of Carbamoyl-Phosphate Synthase 1 (CPS1), which is required for *de novo* biosynthesis of L-citrulline. L-citrulline is essential for *de novo* synthesis of L-arginine, but it is not present in the RPMI growth medium and, thus, its availability relies on *de novo* synthesis pathway via CPS1 and other genes (see S7 Table). CPS1 is a bottleneck in *de novo* synthesis of L-citrulline, being only expressed in intestinal epithelial cells and liver cells. Consequently, L-arginine is essential in most cases under RPMI growth medium unless L-citrulline is additionally supplemented, which was precisely the strategy followed in [18] in order to demonstrate the lethal interaction between L-arginine depletion and loss of ASS1. This insight again reinforces the importance of the tumor microenvironment to predict synthetic lethality.

On the other hand, the dependence on extracellular cholesterol in tumors with loss of squalene monooxygenase (SQLE) was demonstrated in lymphoma [29]. They also showed that the inhibition of the low-density lipoprotein (LDL) receptor (LDLR) was a good proxy for cholesterol depletion. In addition to SQLE, our ngMCS approach in conjunction with RNA-seq data from CCLE [9], allow us to identify other genes implied in cholesterol auxotrophic cell lines: NSDHL, SC5D, FDFT1 and EBP. We identified 18 cell lines whose dependence on extracellular cholesterol was caused by the loss of one of these 5 genes, assuming a threshold of expression of 1 TPM. In agreement with the work of Garcia-Bermudez *et al.*, 2019, 8 out of these 18 cholesterol auxotrophic cell lines are derived from lymphoma; however, we also detected 3 cell lines from endometrial adenocarcinoma (Fig 4B). For comparison, Fig 4B also includes 18 cell lines that are not dependent on extracellular cholesterol (cholesterol prototrophic cell lines), particularly those with the highest expression of the genes involved in the ngMCSs associated with cholesterol. Using large-scale silencing data from DepMap [8], we observed a similar effect of *LDLR* down-regulation in both lymphoma and endometrium cholesterol auxotrophic

a

| | |
|---|---|
| **ngMCS_1:** {CPS1, L-Asparagine} | **ngMCS_6:** {EBP, SLC38A4, Cholesterol} |
| **ngMCS_2:** {GLUL, L-Glutamine} | **ngMCS_7:** {FDFT1, SLC38A4, Cholesterol} |
| **ngMCS_3:** {ASS1, L-Arginine} | **ngMCS_8:** {NSDHL, SLC38A4, Cholesterol} |
| **ngMCS_4:** {CPS1, L-Arginine} | **ngMCS_9:** {SLC38A4, SQLE, Cholesterol} |
| **ngMCS_5:** {ISYNA1, Myo-Inositol} | **ngMCS_10:** {CYP7A1, SLC38A4, SC5D, Cholesterol} |

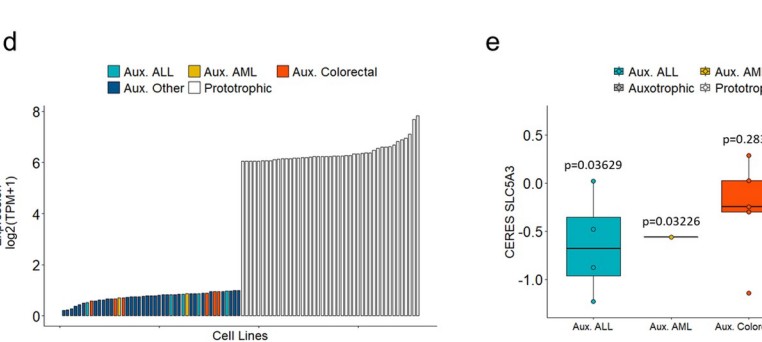

**Fig 4. Extracellular nutrient dependences of *in vitro* cancer cell lines. a)** Relevant ngMCSs involving L-Asparagine, L-Glutamine, L-Arginine, Myo-Inositol and Cholesterol. **b)** Expression levels of the most limiting gene in the ngMCSs associated with Cholesterol in 18 auxotrophic and 18 prototrophic cell lines. Auxotrophic cell lines were defined as those where the most limiting gene has an expression level below 1 TPM. For comparison, we took the same number of prototrophic cell lines; **c)** CERES scores for *LDLR* in the cell lines shown in b). The more negative CERES scores are, the more cellular proliferation will be decreased; **d)** Expression levels of the most limiting gene in the ngMCSs associated with Myo-Inositol in 45 auxotrophic and 45 prototrophic cell lines. **e)** CERES scores for *SLC5A3* in the cell lines shown in d). Abbreviations: 'ALL' and 'AML' refer to Acute Lymphoblastic Leukemia and Acute Myeloid Leukemia, respectively. In addition, 'Aux.' refers to auxotrophic cell lines, *e.g.* Aux. ALL refers to auxotrophic ALL cell lines. Note here that in panels (b)-(d) gene expression data was obtained from CCLE [9] and in panels (c)-(e) CERES scores were obtained from the DepMap platform [8]. The significance of the difference in LDLR and SLC5A3 CERES scores in the auxotrophic groups against the corresponding prototrophic group was assessed via a one-tailed Wilcoxon test.

cancer cell lines, substantially superior than the one found in cholesterol prototrophic cell lines, according to the observed CERES scores (Fig 4C, one-tailed Wilcoxon test p-value = 0.003996 for lymphoma and p-value = 0.06061 for endometrial adenocarcinoma). Note here that CERES is a method to estimate gene dependency from CRISPR-CAS9 loss-of-function screens developed by the DepMap initiative. A gene with more negative CERES score in a particular cell line implies a major depletion of its associated sgRNAs in the population of such cell line, which indicates a higher dependence for cell viability.

In addition, patients with AML and loss of inositol-3-phosphate synthase 1 (*ISYNA1*) have shown a dependency on extracellular myo-Inositol [30]. In this study, the inhibition of the myo-Inositol transporter *SLC5A3* was presented as a good proxy for myo-Inositol depletion.

Again, using RNA-seq data from CCLE, we identified 45 myo-Inositol auxotrophic cell lines, assuming a threshold of expression of 1 TPM for *ISYNA1*. Two of these cell lines are derived from AML; however, 5 and 6 of these cell lines correspond to acute lymphoblastic leukemia (ALL) and colorectal adenocarcinoma (CA) (Fig 4D). For comparison, Fig 4D also includes 45 cell lines that are not dependent on extracellular myo-Inositol, particularly those with the highest expression of *ISYNA1*. Using DepMap data, we observed that the most extreme effect of *SLC5A3* down-regulation was found in ALL cell lines (one-tailed Wilcoxon test p-value = 0.0363); however, we could not demonstrate a similar performance in colorectal cancer (Fig 4E).

For completeness, a similar analysis about cholesterol and myo-Inositol dependencies was done considering all cell lines included in DepMap. We found a modest but statistically significant linear correlation between *SLC5A3* CERES scores and the expression of *ISYNA1*, as well as between *LDLR* CERES score and the expression of the partner gene of cholesterol with the highest expression among different ngMCSs (S3 Fig). In both cases, the distribution of gene expression data suggests a bimodal structure (S4 Fig), in line with our assumption that cell lines respond differently to the depletion of cholesterol and myo-Inositol. Finally, we provide in S8 Table our predictions about which cell lines are dependent on cholesterol, myo-Inositol and the rest of nutrients, following our conservative threshold expression of 1 TPM for the genes involved in ngMCSs.

## Discussion

There is an increasing body of literature evidencing that *in-vivo* resistance to metabolic vulnerabilities identified *in-vitro* could be mediated by alternative pathways driven by nutrients typically not included in standard growth media. This is illustrated here with our study about DHFR inhibition, whose anti-proliferative effect under RPMI growth medium is compensated with the addition of thymidine and hypoxanthine. There are other relevant cases in the literature. For example, it was recently reported that the supplementation of uridine rescues the anti-leukemic effect of dihydroorotate dehydrogenase inhibition [17]. A similar result was found in pancreatic ductal adenocarcinoma, where tumor-associated macrophages release pyrimidines to the extracellular medium which confer resistance to gemcitabine [31]. On the other hand, restricting the availability of certain nutrients for tumor cells is an emergent strategy in cancer research. The dependence on L-asparagine is one paradigmatic example that is used clinically in patients with acute lymphoblastic leukemia. Other ongoing works include cholesterol depletion in lymphoma [29] and myo-Inositol depletion in AML [30], which illustrate that this therapeutic strategy could be exploited in wider settings. All these results together emphasize the importance of systematically considering the nutrient environment to target cancer metabolism via synthetic lethality approaches [32].

Here, we propose a novel family of metabolic synthetic lethal interactions, which include genes but also nutrients in the environment, going beyond existing definitions in the literature. For their calculation, we extend our previously developed computational approach to identify synthetic lethal interactions in cancer metabolism [14]. In particular, we move from genetic Minimal Cut Sets, gMCSs, to nutrient-genetic Minimal Cut Sets, ngMCSs. While gMCSs define minimal subsets of gene knockouts perturbations that lead to cellular death, ngMCSs incorporate extracellular nutrient deprivation as part of the predicted synthetic lethal interactions. The ngMCS approach is a more flexible framework that allows us to predict context-specific genetic and nutritional perturbations that lead to cellular death.

Our computational tool could help to assess previously identified synthetic lethal interactions in more complex environmental scenarios and identify combinatorial therapies that

target alternative metabolic pathways. For example, based on the gMCSs shown in Fig 2, DHFR inhibition could be combined with SLC29A2 inhibition to avoid the uptake of hypoxanthine and disrupt the salvage pathway for purines biosynthesis. Our ngMCS approach could also predict nutrient restriction strategies that strengthen the efficacy of metabolic targets identified *in vitro*. This was illustrated in Fig 2, where restricting the availability of thymidine or hypoxanthine makes more effective the inhibition of DHFR in tumor cells. From another perspective, our approach opens new avenues to systematically identify novel response biomarkers to existing metabolic treatments, beyond frequently used genomic biomarkers [33].

In addition, based on RNA-seq data, the ngMCS approach can be used to predict extracellular nutrient dependences of tumor cells. In our analysis, summarized in Fig 4, we found that the lethality of cholesterol depletion previously reported in lymphoma can be extended to a subgroup of endometrial adenocarcinoma cell lines. Similarly, we pose the relevance of myo-Inositol depletion in acute lymphoblastic leukemia, going beyond previous results in acute myeloid leukemia. Although further work is required to assess the clinical relevance of these results, they illustrate the importance of studying more systematically the role of tumor microenvironment in order to identify metabolic vulnerabilities.

One may argue that targeting nutrient dependencies could largely overlap with the corresponding removal of nutrient transporter. However, transporters are not specific in many cases and could accept several nutrients. Moreover, nutrients typically have different transport systems, which can be more difficult to target. Myo-Inositol and cholesterol dependencies were assessed with one of the transporters previously recognized in the literature as relevant for their uptakes; however, this is not the optimal experiment. The optimal experiment would be to deplete myo-Inositol and cholesterol from the growth medium, as done in other works [18]. As the availability of large-scale studies evaluating the effect of nutrient depletions in cancer cell lines are lacking, the type of validation for our predictions is currently limited.

Current therapeutic approaches to control the level of nutrients in the tumor microenvironment include nutrient-degrading enzymes and custom diets. The intravenous administration of L-asparaginase enzyme to acute lymphoblastic leukemia patients is one successful example. This enzyme, synthetized from *E. coli*, involves relevant toxic effects; however, they are nonfatal, manageable, and reversible [34], which is encouraging for further developments. On the other hand, the effectiveness of custom diets in cancer has been successfully tested in mouse models [35]. However, their application to cancer patients is an open question and further clinical studies are required to demonstrate their benefit.

Finally, the study of tumor microenvironment and nutrient availability requires the use of metabolomic approaches. Availability of metabolomics data in different biofluids is growing day-by-day in cancer studies [36]; however, metabolomics studies in tumor extracellular microenvironment are less frequent but crucial to understand metabolic activity and identify metabolic vulnerabilities [37,38]. Once this information is available for different tumors, our ngMCSs approach constitutes an elegant strategy to integrate genomics, transcriptomics and metabolomic data with genome-scale metabolic networks and predict synthetic lethals and genetic and nutrient dependencies in cancer.

## Methods

### Metabolic networks and synthetic lethality

For the results presented above, we used Recon3D_3.01 as reference human genome-scale metabolic network [15], which is available in https://www.vmh.life/. Recon3D_3.01 involves 13,543 reactions, where collectively participate 4,138 metabolites and 3,695 genes. Note here that we corrected annotation errors previously identified in Recon 2 [14,39] and inherited in

Recon3D_3.01. In addition, we deleted HMR_9797 reaction, since adenine deaminase function has not been reported in human cells [40]. All corrections are summarized in S9 Table.

In the Results section, we used three different growth medium conditions. First, we simulated the RPMI1640 culture medium following the nutrient availability provided by the formulation of RPMI1640 with L-Glutamine (Lonza, Basel, Switzerland), similar to the one reported in [41]. However, for completeness, we also included in our RPMI formulation the nutrients provided by fetal bovine serum (FBS), typically added to *in-vitro* cell cultures (see S3 and S4 Tables for details). Second, we simulated the most general growth medium conditions by enabling all input exchange fluxes of nutrients available in Recon3D_3.01. Finally, we simulated the RPMI1640 culture medium, as described above, plus thymidine and hypoxanthine, as discussed in Figs 2 and 3.

Genome-scale metabolic networks can be used to predict metabolic synthetic lethals. From a network perspective, a synthetic lethal is defined as a subset of genes whose simultaneous loss blocks the flux through the biomass reaction. This reaction integrates the essential metabolic requirements for cellular proliferation and its associated flux represents the proliferation rate. In our analysis, we used the default biomass reaction available in Recon3D_3.01.

## The genetic Minimal Cut Set approach

The identification of synthetic lethals was done through our previously developed approach termed genetic Minimal Cut Sets (gMCSs) [14]. Given a reference metabolic network for a specific growth medium, gMCSs define minimal subsets of gene whose simultaneous loss disrupt a particular metabolic task. When gMCSs are applied to block the flux through the biomass reaction, gMCSs match with synthetic lethals.

In contrast with other approaches in the literature that first build context-specific metabolic models using -omics data and then identify lethal single gene knockouts [42], we avoid this contextualization step and directly calculate gMCSs from the reference (uncontextualized) metabolic network. Once they are calculated, we integrate -omics data and search for context-specific genetic dependencies of tumors. In particular, a gene is considered essential for a particular context if it is the only expressed element in at least one gMCS. With our gMCSs approach and gene expression data, we showed a superior performance than other algorithms in the literature to predict essential genes in cancer [14].

From a computational perspective, the exhaustive search of gMCSs from the full space of gene deletions is a combinatorial problem, which is infeasible even for networks of moderate size. However, we can enumerate a subset of gMCSs by increasing number of genes via mixed-integer linear programming (MILP) using standard solvers such as IBM ILOG CPLEX (see details in [43]), *i.e.* we calculate the shortest gMCS, then the second shortest, and so on until a termination criterion is satisfied. The computation of the shortest gMCS is NP-hard; however, it is tractable for the available genome-scale metabolic networks of human cells.

In addition, the computation time to solve our MILP approach can be alleviated heuristically by fixing a time limit in the search process at the expense of generating false positives (solutions that are not minimal in terms of number of genes). We previously showed that a time limit per gMCS of 30 seconds restricts the rate of false positives to less than 1% in the computation of 1000 solutions for Recon2.v04 and Recon3D_3.01 [43]. Overall, our MILP approach is able to determine a subset of gMCSs from which we can effectively elucidate genetic dependences of cancer cells.

## The nutrient-genetic Minimal Cut Sets approach

Here, we extend our gMCS approach in order to consider the environmental context and nutrients availability. In particular, we introduce a novel concept: the nutrient-genetic

Minimal Cut Sets (ngMCSs), which are defined as minimal subsets of gene functions and/or nutrients in the environment whose simultaneous loss block a particular metabolic task, here the flux through the biomass reaction. We describe below how our previous formulation is amended to incorporate the loss of nutrients in the environment in the identification of synthetic lethal interactions.

A central part of our gMCSs approach is the construction of the binary $G$ matrix, where each row defines the reactions deleted by a minimal subset of gene knockouts. Then, based on duality theory and mixed-integer linear programming, we can identify minimal combinations of rows of $G$ that blocks the biomass reaction. Full details can be found in [43].

In genome-scale metabolic reconstructions, input exchange reactions represent the availability of different nutrients in the environment. These reactions do not include any genetic association and, consequently, they cannot be blocked through any gene knockout, being their associated columns in $G$ always zero. In order to consider their removal, which enables us to model the deprivation of nutrients in the environment, the $G$ matrix must be amended. In particular, we need a new row for each nutrient in the environment with all entries equal to '0' except for the column associated with its input exchange reaction. With this simple extension of $G$, ngMCSs can be determined using our previously developed algorithms for gMCSs.

In order to amend $G$ matrix for the calculation of ngMCSs, we created an artificial gene for each different input exchange reaction present in the environment, *e.g.* gene_Ex_thymidine for the input exchange reaction of thymidine. This updated metabolic model was introduced as input data to our previously developed MATLAB function to calculate gMCSs (*CalculateGeneMCS*), freely available in the COBRA Toolbox [21], which requires IBM ILOG CPLEX to solve the underlying mixed-integer linear programming models. In this setting, the result could be both gMCSs and ngMCSs. We modified our MATLAB function to allow the user to calculate ngMCSs in the COBRA Toolbox (code available in S1 Code).

For the study of the lethality of DHFR, we followed two different strategies to calculate ngMCSs in the most complex environment with all nutrients present in Recon3D_3.01 available. First, we explored all possible combinations of gene knockouts and nutrient deprivations. Second, combinations of the DHFR knockout and nutrient deprivations were directly analyzed. Note here that *CalculateGeneMCS* function can restrict the search space among a predefined list of genes (*gene_set* optional parameter). The resulting gMCSs and ngMCSs are detailed in S1 and S2 Tables. A similar analysis was done for the environment defined by the nutrients in the RPMI growth medium plus thymidine and hypoxanthine. The list of obtained gMCSs and ngMCSs are shown in Fig 2. For the study of extracellular nutrient dependencies of cancer cell lines, we considered all possible combinations of gene knockouts and deprivations of nutrients available in the RPMI growth medium. The list of ngMCSs can be also found in S7 Table. These results were computed with Intel(R) Xeon(R) Silver 4110 CPU @ 2.10GHz processors, limiting to 8 cores and 8 GB of RAM. A time limit of 60 seconds was set for each solution derived from the function *CalculateGeneMCS*.

## Computational modelling of DHFR knockout

The knockout of DHFR leads to the accumulation of DHF (dihydrofolate), which blocks the enzyme AICART (aminoimidazole carboxamide ribonucleotide transformylase) [44]. At the same time, the inhibition of AICART increases the levels of 5-aminoimidazole-4-carboxamide-1-β-d-ribofuranosyl 5′-monophosphate (AICAR), which inhibits adenosine deaminase (ADA) and AMP deaminase (AMPDA) [44]. In summary, in our presented analysis, the knockout of DHFR involves the removal of their associated reactions in Recon3D_3.01 and, indirectly, the reactions associated with AICART, ADA and AMPDA.

## Computational modelling of extracellular nutrient dependences in cancer cell lines

We neglected 5 reactions in Recon3D_3.01 annotated to protein degradation, since the metabolism of macromolecules is not well represented, leading to unbalanced and meaningless cycles. For consistency, we also deleted the availability of albumin from our simulations under the RPMI growth medium. This assumption does not affect the main results obtained for cholesterol and myo-Inositol.

## Cell culture

The cell lines PF-382 and JVM-2 were maintained in culture in RPMI1640 medium (Gibco, Grand Island, NY) and HT-29 cells with McCoy's 5a medium (Gibco, Grand Island, NY), all of them supplemented with 10% fetal bovine serum (Gibco, Grand Island, NY) and penicillin/streptomycin (BioWhitaker, Walkersvill, MD) at 37˚C in a humid atmosphere containing 5% $CO_2$. Cell lines were obtained from the DSMZ or the American Type Culture Collection (ATCC). All cell lines were authenticated by performing a short tandem repeat allele profile and were tested for mycoplasma (MycoAlert Sample Kit, Cambrex), obtaining no positive results.

## Cell proliferation assay

Cell proliferation was analyzed using the CellTiter 96 Aqueous One Solution Cell Proliferation Assay (Promega, Madison, W). This is a colorimetric method for determining the number of viable cells in proliferation. For the assay, cells were cultured by triplicate in 96-well plates, PF-382 at a density of $1 \times 10^6$ cells/mL (100.000 cells/well, 100μL/well) and JVM-2 at a density of $2 \times 10^5$ cells/mL (20.000 cells/well, 100μL/well). HT-29 cells were obtained from 80–90% confluent flasks and 100 μL of cells were seeded at a density of 5000 cells /well in 96-well plates by triplicate. Before addition of the compound, adherent cells were allowed to attach to the bottom of the wells for 12 hours. In all cases, only the 60 inner wells were used to avoid any border effects.

After 96 hours of treatment with different doses of methotrexate (MTX) (Selleckchem, TX, USA), plates with suspension cells were centrifuged at 800 g for 10 minutes and medium was removed. The plates with adherent cells were flicked to remove medium. Then, cells were incubated with 100 μL/well of medium and 20 μL/well of CellTiter 96 Aqueous One Solution reagent. After 1–3 hours of incubation at 37˚C, the plates were incubated for 1–4 hours, depending on the cell line at 37˚C in a humidified, 5% $CO_2$ atmosphere. The absorbance was recorded at 490 nm and 640 nm as a reference wavelength, using 96-well plate readers until absorbance of control cells without treatment was around 0.8. The background absorbance was measured in wells with only cell line medium and solution reagent. First, the average of the absorbance from the control wells was subtracted from all other absorbance values. Data were calculated as the percentage of total absorbance of treated cells/absorbance of non-treated cells. The $GI_{50}$ values of the different compounds were determined using non-linear regression plots with the GraphPad Prism v5 software.

## MTX treatment

PF-382, JVM-2 and HT-29 cells were seeded by triplicate in 96-well plates at 100000, 20000 and 5000 cells per well. HT-29 cells were plated 24 hours before treatment. Then, all cell lines were treated with 20, 50, 75 and 100nM of MTX (Selleckchem, TX, USA) in presence or absence of 100μM hypoxanthine and 16μM thymidine (HT) (ThermoFisher Scientific). The

cell proliferation assay was performed 96 hours after as described above. First, the average of the absorbance from the control wells was subtracted from all other absorbance values. Data were calculated as the percentage of total absorbance of treated cells/absorbance of non-treated cells.

## Cell transfection

Cells were passaged 24 h before nucleofection and culture was divided into two parts. One continues growing under same conditions (absence of HT) while the other was supplemented with 100μM hypoxanthine and 16μM thymidine. The transfection of siRNAs was done with the Nucleofector II device (Amaxa GmbH, Köln, Germany) following the Amaxa guidelines. Briefly, $1 \times 10^6$ of PF-382 and JVM-2 cells were resuspended in 100 μL of supplemented culture medium, with or without HT, with 75 nM of *DHFR* siRNAs or Silencer Select Negative Control-1 siRNA (Ambion, Austin, TX) and nucleofected with the Amaxa nucleofector apparatus using programs C-009 and C-006, respectively. In the case of HT-29, siRNAs were transfected using lipofectamine transfection reagent 2000 (Invitrogen, Carlsbad, CA) according to manufacturer's protocol. Briefly, HT-29 cells (50,000 cells per well) were seeded in a six-well plate with antibiotic-free medium 24 h before transfection. Cells were then incubated with transfection mixtures containing 75 nM of siRNAs or Silencer Select Negative Control-1 siRNA (Ambion, Austin, TX) for 4 h. Then, medium was replaced with full culture medium. Transfection efficiency was determined by flow cytometry using the BLOCK IT Fluorescent Oligo (Invitrogen Life Technologies, Paisley, UK). We used four different siRNAs against *DHFR* target, two for the isoforms 1, 3 and 4 and another two for the isoform 2 (siDHFR-134 A: AAGUCUAGAUGAUGCCUUA; siDHFR-134 B: AACCAGAAUUAGCAAAUAA; siDHFR-2 A: AGUACAAAUUUGAAGUAUA; siDHFR-2 B: AAAUUGAUUUGGAG AAAUA) to demonstrate that the results obtained with *DHFR* siRNA nucleofection are not due to a combination of inconsistent silencing and sequence specific off-target effects. Silencer Select Negative Control-1 siRNA was used to demonstrate that the nucleofection did not induce non-specific effects on gene expression. Nucleofection was performed twice with a 24 h interval. After 48 h of the second nucleofection, the *DHFR* mRNA expression was analyzed by qRT-PCR (*GUS* was employed as the reference gene). Cell proliferation was analyzed 0, 2, 4 and 6 days after two repetitive transfections as described above. HT were refreshed every two days. First, the average of the absorbance from the control wells was subtracted from all other absorbance values. Data were calculated as the percentage of total absorbance of *DHFR* transfected cells/absorbance of control cells.

## Quantitative RT-PCR

The expression of *DHFR* was analyzed by qRT-PCR in PF-382, JVM-2 and HT-29 cell lines. First, total mRNA was extracted with Trizol Reagent 5791 (Life Technologies, Carlsbad, CA, USA) following the manufacturer instructions. RNA concentration was quantified using NanoDrop Specthophotometer (NanoDrop Technologies, USA). cDNA was synthesized from 1 μg of total RNA using the PrimeScript RT reagent kit (Perfect Real Time) (cat. no. RR037A, TaKaRa) following the manufacturer's instructions. The quality of cDNA was checked by a multiplex PCR that amplifies *PBGD*, *ABL*, *BCR*, and *β2-MG* genes. qRT-PCR was performed in a 7300 Real-Time PCR System (Applied Biosystems), using 20 ng of cDNA in 2 μL, 1 μL of each primer at 5 μM (DHFR F:5′-CCATTCCTGAGAAGAATCGAC-3′; DHFR R:5′- GGCAT-CATCTAGACTTCTGGAAA-3′; GUS F: 5′-GAAAATATGTGGTTGGAGAGCTCATT-3′; GUS R:5′-CCGAGTGAAGATCCCCTTTTTA-3′), 6 μL of SYBR Green PCR Master Mix 2X (cat. no. 4334973, Applied Biosystems) in 12 μL reaction volume. The following program

conditions were applied for qRT-PCR running: 50˚C for 2 min, 95˚C for 60 s following by 45 cycles at 95˚C for 15 s and 60˚C for 60 s; melting program, one cycle at 95˚C for 15 s, 40˚C for 60 s and 95˚C for 15 s. The relative expression of each gene was quantified by the Log2 (−ΔΔCt) method using the gene *GUS* as an endogenous control.

## Supporting information

**S1 Code. ZIP file compressing the MATLAB code for computing ngMCSs, the R code for generating Figs 4 and S1–S4 and necessary input data.**
(ZIP)

**S1 Table. List of identified gMCSs involving DHFR with all nutrients present in Recon3D available in the growth medium.** Gene Symbol and Entrez identifiers are provided for genes.
(XLSX)

**S2 Table. List of identified ngMCSs involving DHFR will all nutrients present in Recon3D available in the growth medium.** Gene Symbol and Entrez identifiers are provided for genes, while Recon3D identifiers for nutrients (metabolites). ngMCSs shaded in yellow are mentioned in the main text.
(XLSX)

**S3 Table. Composition of the RPMI Culture media used in the experiments and its correspondence to Recon3D metabolites.** Input flux is limited to millimolar concentration of the metabolite available in the culture media.
(XLSX)

**S4 Table. Composition of the FBS supplement used in the experiments and its correspondence to Recon3D metabolites.** Input flux is limited to millimolar concentration of the metabolite available in the culture media.
(XLSX)

**S5 Table. Composition of the McCoy's 5A Culture media used in the experiments and its correspondence to Recon3D metabolites.** Input flux is limited to millimolar concentration of the metabolite available in the culture media.
(XLSX)

**S6 Table. Expression data in TPM for genes implied in the gMCSs and ngMCSs shown in Fig 2 in the cell lines tested in Fig 3.** They were derived from Cancer Cell Line Encyclopedia.
(XLSX)

**S7 Table. List of identified ngMCSs present in Recon3D available under RPMI growth medium computed for searching for extracellular nutrient dependencies.** Gene Symbol and Entrez identifiers are provided for genes, while Recon3D identifiers for nutrients (metabolites). ngMCSs shaded in yellow are mentioned in the main text.
(XLSX)

**S8 Table. Binary matrix showing the essentiality (1) / non-essentiality (0) of the nutrients present in the RPMI culture medium across the CCLE cell lines assuming a threshold of expression of 1 TPM for genes involved in ngMCS.**
(XLSX)

**S9 Table. Corrections performed to Recon3D metabolic network.**
(XLSX)

**S1 Fig. Proliferation of PF-382, JVM-2 and HT-29 cell lines nucleofected with siRNAs targeted to DHFR gene in presence or absence of Hypoxanthine-Thymidine was studied by MTS at day 2, 4 and 6 after nucleofection.** The proliferation percentage refers to cells nucleofected with a negative control siRNA. Data represent mean ± standard deviation of four experiments.
(TIFF)

**S2 Fig.** Expression level of the limiting gene in the ngMCSs involving **a)** L-tyrosine, **b)** glucose and **c)** choline in cancer cell lines obtained from CCLE (Ghandi et al., 2019).
(TIF)

**S3 Fig. a)** Dependence of the CERES essentiality score of LDLR on the expression of the limiting gene in the ngMCSs involving cholesterol. The slope of the regression line is statistically significant (p = 0.00464, r = 0.097). **b)** Dependence of the CERES essentiality score of SLC5A3 on the expression of the limiting gene in the ngMCSs involving myo-Inositol. The slope of the regression line is statistically significant (p = 6.67·10–6, r = 0.154). Gene expression data was obtained from CCLE (Ghandi et al., 2019) and CERES scores were obtained from the DepMap platform (Tsherniak et al., 2017).
(TIF)

**S4 Fig.** Expression level of the limiting gene in the ngMCSs involving **a)** cholesterol and **b)** myo-Inositol in cancer cell lines obtained from CCLE (Ghandi et al., 2019).
(TIF)

## Author Contributions

**Conceptualization:** Iñigo Apaolaza, Edurne San José-Enériz, Luis V. Valcarcel, Xabier Agirre, Felipe Prosper, Francisco J. Planes.

**Data curation:** Iñigo Apaolaza, Luis V. Valcarcel.

**Funding acquisition:** Felipe Prosper, Francisco J. Planes.

**Investigation:** Iñigo Apaolaza, Edurne San José-Enériz, Luis V. Valcarcel, Xabier Agirre, Felipe Prosper, Francisco J. Planes.

**Methodology:** Iñigo Apaolaza, Luis V. Valcarcel, Francisco J. Planes.

**Software:** Iñigo Apaolaza, Luis V. Valcarcel.

**Supervision:** Xabier Agirre, Felipe Prosper, Francisco J. Planes.

**Validation:** Edurne San José-Enériz, Xabier Agirre.

**Visualization:** Iñigo Apaolaza, Francisco J. Planes.

**Writing – original draft:** Iñigo Apaolaza, Francisco J. Planes.

**Writing – review & editing:** Iñigo Apaolaza, Edurne San José-Enériz, Luis V. Valcarcel, Xabier Agirre, Felipe Prosper, Francisco J. Planes.

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
