## [Decision Letter · Decision Letter 0]

16 Oct 2021

Dear Professor Planes,

Thank you very much for submitting your manuscript "A network-based approach to integrate nutrient environment in the prediction of synthetic lethality in cancer metabolism" for consideration at PLOS Computational Biology.

As with all papers reviewed by the journal, your manuscript was reviewed by members of the editorial board and by several independent reviewers. In light of the reviews (below this email), we would like to invite the resubmission of a significantly-revised version that takes into account the reviewers' comments.

We cannot make any decision about publication until we have seen the revised manuscript and your response to the reviewers' comments. Your revised manuscript is also likely to be sent to reviewers for further evaluation.

Sincerely,

Nathan E Lewis

Guest Editor

PLOS Computational Biology

Kiran Patil

Deputy Editor

PLOS Computational Biology

Reviewer's Responses to Questions

**Comments to the Authors:**

Reviewer #1: The authors have conducted an in silico study of synthetic lethality that adds nutrient availability as a dependency alongside genes. They validate one of their predictions in vitro and use published high throughput in vitro data to corroborate two additional predictions. The computational approach is a natural extension of previous work and can be expected to overlap with results involving transporter knock outs to some degree, but the method should still be of interest given the importance of the tumor micro environment for understanding differential responses to therapies. The clarity and discussion of clinical feasibility/study limitation could be further improved in a minor revision as detailed in the following specific comments.

Major comments:

The authors propose a new definition of synthetic lethality; however, it is not clear that a new definition is warranted. A downside with the proposed definition is that two different entities (genes and metabolites) are included under the same concept, although this may perhaps be alleviated by considering them druggable dependencies. The definition proposed in the abstract is not sufficiently precise, and it may perhaps be premature to propose an expansion of an existing definition in an original research article. Perhaps context dependent lethality/SL will suffice for now?

A stronger case for the clinical relevance of context dependent SL could be made in the introduction, e.g. it is mentions that SL is a promising approach, but not how; it is mentioned that cancer could be an application of SL but it is not established that nutrient availability could be a druggable target, although in the result section it is later mentioned that depletion of certain amino acids have been used as therapy. The reverse concept of nutritional requirements to overcome lethality would also benefit from a more through introduction. How do the authors envision that nutrients could be controlled in the micro environment? Or if depletions are systemic, what side-effects could be anticipated and tolerable?

The reader would benefit from a more thorough description of the cut set method in the background. Integer programming is an NP-complete problem, so is the method guaranteed to find all solutions, or will it only find a subset? The method section mentions a 60 second search cut-off, if this was increased by several orders of magnitude, would more and/or different cut sets be expected? For linear pathways where all reactions have the same effect, which one will be selected by the model? Will this choice be consistent or stochastic? Why are gMCS of size 1 discarded? How does the gMCS method compare to other approaches for SL using genome scale models, e.g. Agren 2014.

The method enables identification of more simultaneous knockouts than brute force methods, but is it reasonable to expect that 10+ dependencies could be targeted clinically? Perhaps it is reasonable to say that the window of relevance for this method lies between around 3-5 KOs, i.e. where brute force is infeasible (>2 dependencies) yet the results are clinical is feasible (<~5). Would it be feasible to introduce an upper bound on number of elements in the cut-set? Alongside the elegant case studies performed in this paper, it seems like the authors should be in a good position to perform a more global analysis, how many clinically feasible SL options are there? How many metabolites are in these sets? Are the 8 nutrients that were identified the only ones that could be SL? Could the analysis in figure 4 be done more systematically? e.g. scattering expression of dependencies against CERES scores across multiple conditions?

The result of nutrient availability can be expected to largely overlap with KO of the corresponding nutrient transporter, this is even the premises for figure 2, it may be useful to discuss when the method can be expected to give a different result and/or if this is mostly for analytical convenience.

A more through discussion on the requirements and assumptions of the method would be useful. The method depends on a static biomass equation. But in the living cell biomass is not static. This can lead to situations where a reaction is computationally identified as essential, but not necessarily biologically essential, e.g. glycogen is an energy storage molecule that cells likely can do without, e.g. some cancer cell lines have undetectable glycogen levels while others have elevated levels (Yates 2016). The method mentions different curation efforts to the model, are these anticipated to affect the results? If so it would be appropriate to mention that they have been performed in the main text and discuss in which way the results may be affected? It would perhaps also be relevant to reproduce the results in another GEM, e.g. Human1 (Robinson 2020), this model encourages users to report annotation errors and regularly updates the model. The minimal cut sets the method could be expected to disfavor knocking out reactions with multiple isozymes over knocking out two completely different reactions, however, it is conceivable that a single drug would block all isozymes (as is the case for MTX in this study).

A discussion of the assumption about growth medium may also be warranted. In this study the growth medium was modeled as the components of RPMI, however the cell cultures are grown with 10% FBS that contains a large number of metabolites, could this be expected to influence the results? Additionally, the HT-29 cells were grown in McCoy medium, it may be good to mention if there are any differences in composition between these media that may be of relevance, in particular since HT-29 cells were observed to react differently. In this study degradation of macromolecules is neglected for computational convenience, but could it be biologically feasible for cells to meet their amino acid needs by protein uptake and subsequent degradation?

The authors use a cutoff of 1 TPM to consider a gene active. While this is a common practice, there are many intermediate steps between gene expression and flux, i.e. lower expression does not imply lower flux when comparing two different genes. The relative gene expression between cell lines and contexts may sometimes be more relevant to gauge activity. The authors discuss “the most limiting gene” but it is not guaranteed that the gene with lowest expression is most limiting. For L-tyrosine the text mentions lowly expressed ngMCS, however, no data is provided to support this claim. The authors report expression data in figure 4 that suggest a bimodal distribution of expression, but cell lines with intermediate expression have been removed. It is of interest to see the distribution of the intermediate genes, such information can be provided as supplementary material.

If a claim is made about difference between conditions, e.g. in figure 3 and 4, then it is convention to perform a statistical test on the data, e.g. and show the p value.

Minor comments:

The abstract makes claims of novelty. This is often considered redundant in original research articles.

The abstract mentions synthetic lethality (SL), but it is not explained.

The “tumor environment”, is more commonly referenced to as “tumor micro environment”.

Order of tabs in supplementary data 2 is different than in the index.

Abbreviation “ngMCSs” used in Figure 1 before being defined.

Due to different international standards, it is common to give the base of the logarithm in subscript (e.g. Log10) if it is not the natural logarithm (e.g. figure 3).

The source of data should be made clear in the legend of figure 4.

The experimental basis of the CERES score could be explained more clearly.

According to methods proliferation was measured at day 0, 2, 4, and 6, but only day 6 is reported in figure 3c, this data should perhaps be made available as supplementary data.

The literature review of DHFR knock out in the methods section is not a method and should perhaps be moved to introduction or results.

Lethality is taken to mean a blocked biomass reaction, but method also mentions “removal disrupts the flux through the biomass reaction”, which is weaker than blocking, if there is a distinction it should be clarified.

Overall the language is acceptable, however several instances of reversed word order suggest that the manuscript could benefit from proofreading by a native speaker.

References

Agren, R., Mardinoglu, A., Asplund, A., Kampf, C., Uhlen, M., & Nielsen, J. (2014). Identification of anticancer drugs for hepatocellular carcinoma through personalized genome-scale metabolic modeling. Molecular systems biology, 10(3), 721.

Abstract 1040: Differential levels of glycogen in breast cancer cell lines: A potential new target Joel A. Yates, Megan Altemus, Zhifen Wu, Michelle L. Wynn and Sofia D. Merajver Cancer Res July 15 2016 (76) (14 Supplement) 1040;

Robinson JL, Kocabaş P, Wang H, Cholley PE, Cook D, Nilsson A, Anton M, Ferreira R, Domenzain I, Billa V, Limeta A, Hedin A, Gustafsson J, Kerkhoven EJ, Svensson LT, Palsson BO, Mardinoglu A, Hansson L, Uhlén M, Nielsen J. An atlas of human metabolism. Sci Signal. 2020 Mar 24;13(624):eaaz1482.

Reviewer #2: My review has been uploaded as a pdf attachment.

Reviewer #3: Apaolaza et al presented a computational biology application to predict synthetic lethal by utilizing the gMCSs identification method they have previously developed. Prediction of synthetic lethality is highly important for cancer treatment, which is very appreciate. This manuscript is based on rational computational consideration and supported by in vitro experiments. However, I have a few major concerns:

1) synthetic lethal should be computed in disease specific context. The full RECON3D metabolic network contains all potential metabolic reactions, but not all of them exist in one disease condition. As the authors described in Fig 1, different cancers may have different metabolic variations that may lead to varied synthetic lethal. It is unclear in computing the lethality of DHFR inhibition, which disease context were considered. For the three validation cell lines, why their gene expression or mutation data were not utilized in the computation. In addition, even there is limited novelty of the gMCSs detection in this work, the computational approach including input data should be detailed.

2) A rigorous computational definition of ngMCSs is needed.

3) It is unclear how gene expression data were utilized in the computation of ngMCSs and if the method considers metabolic flux.

4) Figure legends are too small.

**Have the authors made all data and (if applicable) computational code underlying the findings in their manuscript fully available?**

Reviewer #1: **No: **According to methods proliferation was measured at day 0, 2, 4, and 6, but only day 6 is reported in figure 3c, this data should perhaps be made available as supplementary data.

Reviewer #2: Yes

Reviewer #3: Yes

PLOS authors have the option to publish the peer review history of their article (what does this mean?). If published, this will include your full peer review and any attached files.

Reviewer #1: No

Reviewer #2: **Yes: **Dr. Yiping Wang, Postdoctoral Associate, Benjamin Izar Lab, Columbia University Medical Center

Reviewer #3: No
---

## [Decision Letter · Decision Letter 1]

17 Jan 2022

Dear Professor Planes,

Thank you very much for submitting your manuscript "A network-based approach to integrate nutrient environment in the prediction of synthetic lethality in cancer metabolism" for consideration at PLOS Computational Biology. As with all papers reviewed by the journal, your manuscript was reviewed by members of the editorial board and by several independent reviewers. The reviewers appreciated the attention to an important topic. Based on the reviews, we are likely to accept this manuscript for publication, providing that you modify the manuscript according to the review recommendations.

Sincerely,

Nathan E Lewis

Guest Editor

PLOS Computational Biology

Kiran Patil

Deputy Editor

PLOS Computational Biology

[LINK]

Reviewer's Responses to Questions

**Comments to the Authors:**

Reviewer #1: With this revision most of the comments have been satisfactorily addressed. A few points require further clarification.

Major comments

1) The effect size in supplementary figure 3 looks small based on visual inspection. For a more quantitative argument it would be helpful to also include the correlation coefficient. A low correlation across all expression levels does not detract from the main result, that there are different groups of cell lines that react differently, but if the correlation is low, it would be appropriate to qualify the comments about significant linear relations.

2) Supplementary Figure 4, is used as support for a claim of bimodality. For S4a there does indeed seem to be a bimodal distribution such that values < log2(1+1)=1 belong to one distributions and values > 1 belong to another. However, for S4b this is less clear, although the distribution may perhaps be perceived as two overlapping normal distributions with peaks around 3 and 5.5. Here there is no clear distinction for values <1, which may perhaps help explain the lack of significant difference seen between auxotrophic prototrophic in this setting (with exception for the ALL and AML cell lines).

3) The ngMCSs method can be used to predict essentiality of nutrients in cell lines under the assumption that gene expression below a certain value implies lack of flux for its associated reactions. The authors have already done all the work required to perform a global analysis of nutrient essentiality in RPMI culture medium across all cell lines in the CCLE, which would be a binary matrix [nutrients] x [cell lines]. Such a global analysis would help emphasize the generality of the method and additionally may be a useful resource for experimentalist working with these cell lines.

Minor comments

1) The authors comment that “gMCSs of size 1 were not considered in this study” but for the ngMCSs in RPMI culture medium, the first 9 are of size 1 after the revision. The inclusion seems appropriate for completion, but is there a reason for this difference in definition?

2) For supplementary figure 2 and 4 it would be useful to indicate unit for x axis (log2 tpm+1) and y axis (probability density?)

3) The abbreviation ALL for acute lymphoblastic leukemia is explained in the main text but could perhaps also be spelled out in the figure legend, since it can easily be mistaken to mean all cell lines.

4) On page 24 it is written “One continu. One continues”.

Reviewer #2: I am overall satisfied with the revisions that you have made to the manuscript. The only thing I would still advise that you do is to look more closely at visualizing your results on a pathway map. I still believe this may provide some novel insights, even if it must be restricted only to reactions with a simple reaction-gene relationship. Thank you for your hard work, and good luck with publication.

Reviewer #3: The authors has solved all my questions.

**Have the authors made all data and (if applicable) computational code underlying the findings in their manuscript fully available?**

Reviewer #1: Yes

Reviewer #2: Yes

Reviewer #3: **No: **I could not find the detailed analysis codes. Only example codes (MATLAB) were provided.

PLOS authors have the option to publish the peer review history of their article (what does this mean?). If published, this will include your full peer review and any attached files.

Reviewer #1: No

Reviewer #2: **Yes: **Yiping Wang

Reviewer #3: **Yes: **Chi Zhang

Figure Files:

Data Requirements:

Reproducibility:

References:

---

## [Editor Report · Decision Letter 2]

18 Feb 2022

Dear Professor Planes,

We are pleased to inform you that your manuscript 'A network-based approach to integrate nutrient microenvironment in the prediction of synthetic lethality in cancer metabolism' has been provisionally accepted for publication in PLOS Computational Biology.

Best regards,

Nathan E Lewis

Guest Editor

PLOS Computational Biology

Kiran Patil

Deputy Editor

PLOS Computational Biology

---

## [Editor Report · Acceptance letter]

10 Mar 2022

PCOMPBIOL-D-21-01575R2 

A network-based approach to integrate nutrient microenvironment in the prediction of synthetic lethality in cancer metabolism

Dear Dr Planes,

I am pleased to inform you that your manuscript has been formally accepted for publication in PLOS Computational Biology. Your manuscript is now with our production department and you will be notified of the publication date in due course.

With kind regards,

Katalin Szabo
